# Large Scene Synthesis Controlled With Detailed Text Using View-wise Conditional Joint Diffusion With Hierarchical Spatial Controls

## Abstract

Recently, text-driven large scene image synthesis has made significant progress with diffusion models, but controlling it is challenging. While using additional segmentation map with corresponding texts has greatly improved the controllability of large scene synthesis, adding more texts for large scene generation to faithfully reflect detailed text descriptions is challenging. Here, we propose DetText2Scene, a novel detailed-text-driven large-scale image synthesis with high faithfulness, high controllability with high naturalness in global context for the given descriptions. Our DetText2Scene consists of 1) a hierarchical keypoint-box layout conversion from the detailed text by leveraging large language model for spatial controls, 2) a view-wise conditioned joint diffusion process to synthesize a large scene from the given detailed text and the spatial controls in grounded hierarchical keypoint-box layout and 3) a pixel perturbation-based hierarchical enhancement to hierarchically refine it for global coherence. In experiments, our DetText2Scene significantly outperforms prior arts in text-to-image synthesis with the detailed text as well as our generated keypoint-box layouts qualitatively and quantitatively, achieving strong faithfulness with detailed descriptions, superior controllability, and excellent naturalness in global context in CLIP scores and/or user studies.

## 1 Introduction

Recent advances in text-to-image generation with diffusion models trained with billions of images offer advantages not only in producing high-quality, realistic images (Ramesh et al., 2022; Midjourney; Rombach et al., 2022), but also in applying to various tasks such as image inpainting (Lugmayr et al., 2022; Nichol et al., 2021; Saharia et al., 2022), image editing (Meng et al., 2021; Kim et al., 2022; Hertz et al., 2022; Brooks et al., 2023; Avrahami et al., 2023a;a; Nichol et al., 2021), and image deblurring (Whang et al., 2022; Chung et al., 2022), 3D contents generation (Lin et al., 2023; Tang et al., 2023; Xu et al., 2023; Kim et al., 2023a; Kim & Chun, 2023; Seo et al., 2023) with few-shot adaptation (Kumari et al., 2023; Gal et al., 2022; Ruiz et al., 2022). In particular, some of the recent works in text-to-image synthesis has extended the versatility of the pretrained text-to-image diffusion models to generate large-scale images without computationally expensive additional training on them (Avrahami et al., 2023a; Bar-Tal et al., 2023; Lee et al., 2023).

Synthesizing a large scene from text is still challenging in text-to-image generation. Joint diffusion-based methods were proposed for generating seamless montage of images by performing the reverse generative process across multiple views simultaneously while averaging the intermediate noisy images in the overlapped regions every steps from a text prompt (Bar-Tal et al., 2023; Lee et al., 2023). However, considering such a large canvas, the controllability in text-driven large scene generation is a desirable option to have. Additional semantic segmentation input with corresponding texts can offer great controllability to include multiple objects and humans (usually a few) (Bar-Tal et al., 2023), the controllability only with additional texts has not been explored with more objects and humans.

Unfortunately, the detailed text does not seem to work well for the prior arts in text-to-scene generation, struggling to generate large scenes that fully convey the detailed descriptions with the following critical issues, as also illustrated in Figure 1. Image synthesis using publicly available text-to-image diffusion model seems to have *low faithfulness with the detailed texts* with missing objects and

G) The Legend of Zelda landscape, four girls are dancing on the ground. Gr1) two girls are dancing in the left side. P1) a girl, beautiful game character, wearing a red dress. P2) a girl, beautiful game character, wearing traditional dress. O1) a backpack. Gr2) two girls are dancing in the right side. P1) a girl, beautiful game character, wearing traditional dress. P2) a girl, beautiful game character, wearing a yellow dress.

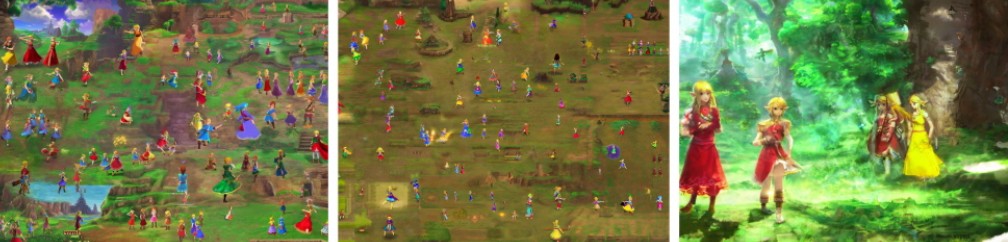

G) The Star Wars Characters on the Alien planet, background is sparkling Milky way and lots of stars. Gr1) the Star Wars Characters on the Alien planet are in the left side. P1, P2, P3) a stormtrooper attacking people. Gr2) The Darth Vader with a light saber in the middle and front side. P1) A Darth Vader, handsome, holding a lightsaber, highly detailed. O1) a lightsaber. GR3) The Star Wars Characters on the Alien planet in the right side. P1, P2, P3, P4) a stormtrooper attacking people.

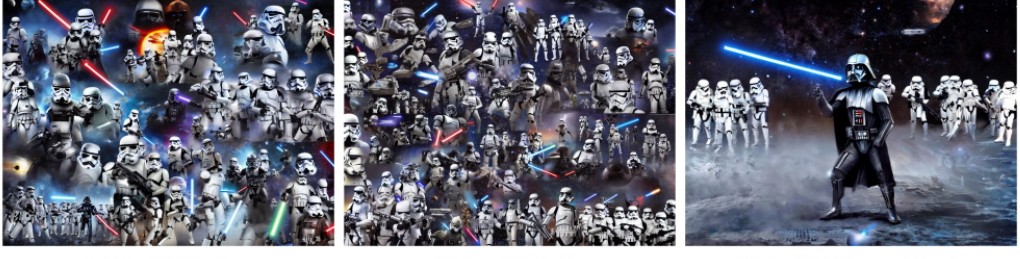

(a) MultiDiffusion          (b) SyncDiffusion          (c) DetText2Scene (Ours)

Figure 1: Examples of detailed text-to-large scene synthesis (2560×1920) from naive applications of prior works, MultiDiffusion (Bar-Tal et al., 2023) and SyncDiffusion (Lee et al., 2023), demonstrating that the prior arts still struggle to generate large scenes from the detailed texts with high faithfulness and controllability with the descriptions and high naturalness in global context, while our proposed DetText2Scene yielded exquisite large scenes from the detailed texts only.

incorrect attribute binding Chefer et al. (2023). For small image synthesis, this issue has been addressed with attention map control Chefer et al. (2023), grounding models Li et al. (2023); Kim et al. (2023b) or additional keypoints (Zhang et al., 2023a), but for large scene synthesis, it has not been yet. Moreover, text-driven large scene synthesis seems to *lack controllability with detailed texts* such as no control on the number of generated humans and objects. One explanation is that joint diffusion process trained on small images (e.g., 512×512) uses the same text prompt for all views, thus overly duplicating object and humans and making them tiny. This joint diffusion has weak dependencies across views that are far apart, which explains *poor naturalness in global context*, yielding disconnected scenes. Additional information like segmentation map with grounded text descriptions to it (Bar-Tal et al., 2023) seems to control well, but controlling large scene synthesis with texts only has been under-explored.

Here, we propose DetText2Scene (Detailed Text To Scene), a novel text-driven large scene synthesis method from detailed texts only to yield *controllable* and *natural* large-scale images *faithfully* corresponding to the texts. Our DetText2Scene consists of 1) grounded hierarchical keypoint and bounding box layout generation from detailed texts by leveraging large language model (LLM) to easily outline several people and objects (see Figure 2, Stage 1), 2) view-wise conditioned joint diffusion process (VCJD) for large scene synthesis, conditioned on spatial controls and ground attributes from generated hierarchical keypoint-box layout and detailed texts (see Figure 2, Stage 2) and 3) pixel perturbation-based hierarchical enhancement (PPHE) (see Figure 2, Stage 3) to hierarchically refine it for further improved quality and global consistency across views to overcome weak dependancies across views. Our experiments demonstrated that our DetText2Scene was capable of detailed-text-to-large-scene synthesis with exquisite quality, achieving strong faithfulness with detailed descriptions (as assessed by CLIP score and 66.35% vs. 33.65% in the user study), strong controllability (as measured by human count accuracy), and high naturalness in both global context (66.35% vs. 33.65% in the user study) and human instance quality (66.35% vs. 33.65% in the user

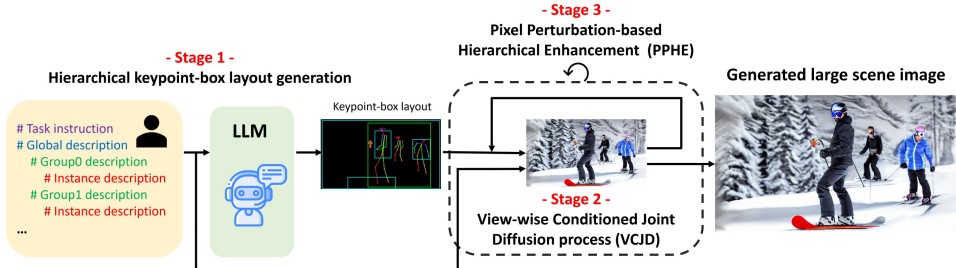

Figure 2: Pipeline of our DetText2Scene, starting from a detailed description, generating a hierarchical keypoint-box layout of the scene from it, synthesizing the target image using our view-wise conditional joint diffusion process with text and spatial controls from keypoint-box layout, and pixel perturbation-based hierarchical enhancement to refine for global coherence across views.

study). Our ablation studies also demonstrate that our DetText2Scene also outperforms prior arts in text-to-scene generation even when keypoint-box layouts were given.

# 2 RELATED WORKS

## 2.1 TEXT-GUIDED LARGE SCENE SYNTHESIS VIA DIFFUSION MODELS

Recent studies in text-guided large scene generation (Bar-Tal et al., 2023; Zhang et al., 2023b; Lee et al., 2023) have simultaneously performed diffusion in multiple views. This involves integrating noisy latent features or scores at each reverse diffusion step. Specifically, when we refer to sampling the subsequent denoised data during the reverse process of diffusion models using the parametrized distribution $p_\theta(\mathbf{x}_{t-1}|\mathbf{x}_t)$ defined in the sampling methods (Ho et al., 2020; Song et al., 2020; Liu et al., 2022) as $\mathbf{x}_{t-1} = \mathcal{D}(\mathbf{x}_t, t, \epsilon)$ where $\epsilon \sim \mathcal{N}(0, \mathcal{I})$, we can consider the case of generating a large-scene image $\mathbf{z} \in \mathbb{R}^{H_z \times W_z \times D}$. Each image within the view, denoted as $\mathbf{x}^{(i)} \in \mathbb{R}^{H_x \times W_x \times D}$, represents a subsection of the image. The collective span of these windows covers the entirety of the large-scene image. We can denote a binary mask as $m^{(i)} \in [0, 1]^{H_x \times H_x}$ for the subregion in the large-scene image corresponding to the $i$-th view. We can define the function $\mathcal{V}_{\mathbf{z} \to i} : \mathbb{R}^{H_z \times W_z \times D} \to \mathbb{R}^{H_x \times W_x \times D}$ that maps the large-scene image $z$ into the $i$-th view image, while The inverse function $\mathcal{V}_{i \to \mathbf{z}} : \mathbb{R}^{H_x \times W_x \times D} \to \mathbb{R}^{H_z \times W_z \times D}$ fills the region outside of the mask $m_i$ with zeros. In the joint diffusion process, where the reverse process is executed simultaneously for each window, the noisy images from the windows $\mathbf{x}_t^{(i)}$ are initially averaged in the large scene space:

$$\mathbf{z}_t = \frac{\sum_i \mathcal{V}_{i \to \mathbf{z}}(\mathbf{x}_t^{(i)})}{\sum_i m^{(i)}}. \tag{1}$$

These joint diffusion-based methods enable to generate seamless arbitrary-sized images. However, existing methods still struggle to generate large scene images which fully convey the detailed descriptions. In this work, we propose DetText2Scene, advancing the joint diffusion to enable to synthesize the large-scale images with significantly improved faithfulness, controllablity and naturalness in both global context, resolving the issues of the unnecessary duplication of objects and disconnectivity.

## 2.2 TEXT-GUIDED IMAGE SYNTHESIS WITH SPATIAL CONTROL VIA DIFFUSION MODELS

Recently, several works (Avrahami et al., 2023b; Li et al., 2023; Qin et al., 2023; Mou et al., 2023; Zhang et al., 2023a) propose to add spatial controls such as a segmentation map (Avrahami et al., 2023b), depth map Li et al. (2023); Qin et al. (2023); Mou et al. (2023); Zhang et al. (2023a), dense caption Xie et al. (2023a); Kim et al. (2023b) and human keypoint Qin et al. (2023); Mou et al. (2023); Zhang et al. (2023a) to a pre-trained text-to-image diffusion model. For example, the reverse diffusion step for image synthesis guided by human keypoint pose map $k$ and text prompt $y$ will be $\mathbf{x}_{t-1} = \mathcal{D}(\mathbf{x}_t, t, y, k, \epsilon)$. These works enable to achieve more controllable synthesis. However, extension to large scale synthesis including many objects will be more challenging and has been under-explored. In this work, we elongate the virtue of controllable synthesis to large scene synthesis. through the view-wise conditioned joint diffusion process, which apply spatial control as well as text description differently upon each view.

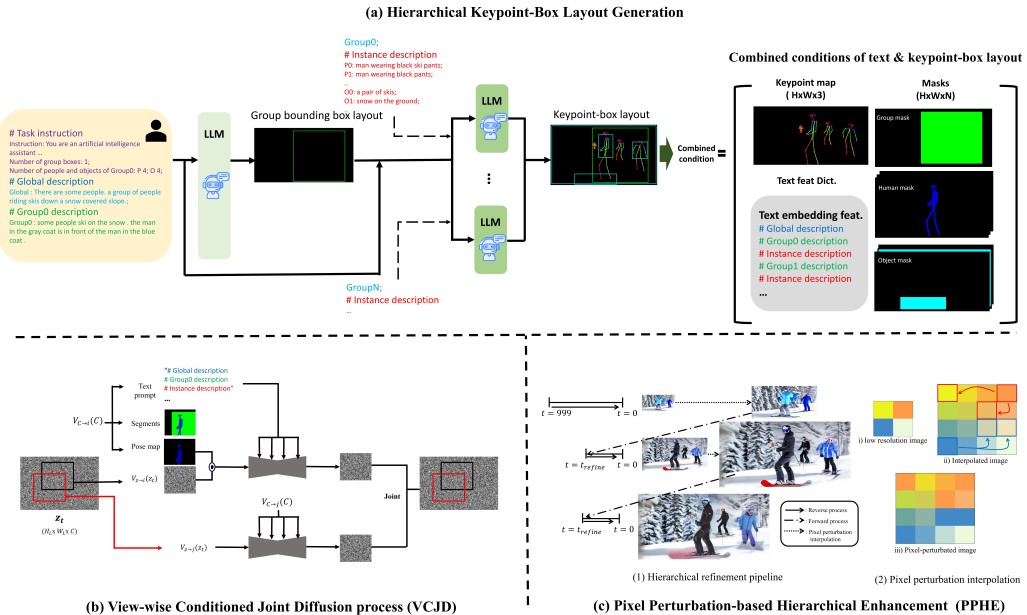

Figure 3: Overview of proposed method. (a) Hierarchical keypoint-box layout generation, (b) View-wise conditioned joint diffusion, (c) Pixel Perturbation-based hierarchical enhancement.

## 2.3 LAYOUT GENERATION VIA LARGE LANGUAGE MODELS

Multiple concurrent studies (Feng et al., 2023; Xie et al., 2023b; Cho et al., 2023) exploit the capabilities of large language models to enhance text-to-image models. LayoutGPT (Feng et al., 2023) utilizes GPT to produce layouts based on text conditions and subsequently generates images from these created layouts. VP-T2I (Cho et al., 2023) fine-tunes an open-source language model specifically for the text-to-layout task and employs standard layout-to-image models for image generation. VisorGPT (Xie et al., 2023b) explicitly learns a probabilistic visual prior through generative pre-training. Nevertheless, methods for generating layouts tailored to large-scene creation from detailed text have been insufficiently explored. To address this, we concurrently generate keypoint-box layouts and employ a hierarchical pipeline to generate complex scenes.

## 3 DETTEXT2SCENE

We first generate a hierarchical keypoint-box layout of the scene, which provides a coarse-to-fine structure for the image generation process, as represented in Fig 3(a). To apply the spatial controls and ground description on each keypoint and bounding box for the large scene, we propose view-wise conditioned joint diffusion process, as represented in Fig 3(b). To ensure the high quality and global dependencies across the images, we propose the pixel perturbation-based enhancement where the diffusion process is repeated from lower size to the target size as represented in Fig 3(c).

### 3.1 HIERARCHICAL KEYPOINT-BOX LAYOUT GENERATION

**Hierarchical keypoint-box layout dataset.** We proposed the hierarchical keypoint-box layout labeled with detailed description based on the CrowdCaption dataset (Wang et al., 2022).As shown in the Fig. 3, the input prompt consists of global, group and instance description to represent the hierarchical structure of the image. To collect proper annotations of our dataset, We firstly extracted the diverse information of dense captions and visual location of each image using existing models of dense captioning (Wu et al., 2022), image captioning (Li et al., 2022) and pose estimation (Xu et al., 2022). Then, we conducted an aligning process of the acquired information to pair each visual location and matched them with the group grounding annotation.

**Instruction tuning for hierarchical keypoint-box layout generation.** As the coarse-to-fine manner, we proposed Hierarchical keypoint-box layout generation and fine-tuned the pre-trained LLM (Touvron et al., 2023) to estimate from global outline to local keypoint-box layout of the image. First, we fine-tuned the global grounding LLM to generate the proper group box layout from the global and group description prompt. Then, the local grounding LLM is fine-tuned to generate the keypoint-box layout of each instance from hierarchical descriptions and generated group box layout prompt.

## 3.2 VIEW-WISE CONDITIONED JOINT DIFFUSION

To apply the spatial controls and ground text description on each keypoint-box for the large scene, we propose view-wise conditioned joint diffusion process (VCJD), which enable to extend Control-Net (Zhang et al., 2023a) and attention modulation to the large scene image generation, as represented in Fig 3(b).

**Global condition set generation.** To use keypoint-box layout for the next step, we convert the raw keypoint and bounding box coordinates to a global condition set $\mathbf{C}$, composed of 1) a pose map, 2) prompt key-grounded masks, and 3) prompt dictionary.

**Dense keypoint-box text-to-image diffusion.** Although keypoint-driven image generation show high quality generation of human instances, grounding on each human keypoint has been under-explored, which will enable detailed control of human instances such as gender, color of clothes and character. To achieve this, we combinated ControlNet-openpose Zhang et al. (2023a) model with grounding method using attention modulation (Kim et al., 2023b). The reverse diffusion step for image synthesis guided by the full-text caption $y$, human keypoint pose map $k$, dense caption $\{(y_n, s_n)\}_{n=1}^{N}$ can be represented as:

$$\mathbf{x}_{t-1} = \mathcal{D}(\mathbf{x}_t, t, y^{(i)}, k^{(i)}, \{(y_n^{(i)}, s_n^{(i)})\}_{n=0\cdots N^{(i)}-1}, \epsilon, \tag{2}$$

where each segment $(y_n, s_n)$ describes a single region, $y_n$ is a non-overlapping description of each human or object that is a part of $y$, and $s_n$ denotes a binary map extracted from box or pose mask layout.

**Large scene image generation with VCJD.** Existing joint diffusion process uses the same text prompt for all views, thus overly duplicating object and humans and making them tiny. Also, extension of conditioning spatial controls to large scale synthesis has been under-explored. Here, we input conditions including text prompt, keypoint-box visual priors to the diffuison models, as represented in Algorithm 1. VCJD enable to yield controllable and natural large-scale images faithfully corresponding to the texts resolving artifacts.

## 3.3 PIXEL PERTURBATION-BASED HIERARCHICAL ENHANCEMENT

Since our method creates a large scene by grounding multiple instances and objects, global consistency must exist for fine quality outputs. To achieve this, we first create a small scene with excellent global consistency, then increasing the size of the scene and refining it. Afterwards, the size of the created image is doubled, and then the forward process is performed in time steps up to $t_m ix$, and then the reverse step is performed again. Afterwards, the image resizing and forward-reverse process are repeated in half of the total time steps until the scene becomes the desired size.

If we just naïvely double the size of the image created in the previous stage, the image becomes blurry, resulting blurry refined outputs. This is because since the blurry image already has a sufficiently high probability density, there is no significant change even after the reverse process. A way to solve this problem is to inject high-frequency components into the image while maintaining semantic information. After interpolating the image, we forcibly increased the high-frequency of the image by swapping two adjacent pixels with a certain probability of $p_{pert}$.

## 4 EXPERIMENT

The same Stable Diffusion 1.5 model (Rombach et al., 2022) is leveraged for all the methods for evaluation. For instruction tuning, we utilized the open-source toolkit of training LLMs (Gao et al.,

---

**Algorithm 1:** Large scene genreation with view-wise conditioned joint diffusion process (VCJD)

---

**Inputs:** $\mathbf{z}_T, \mathbf{C}$;                    // random noise, global condition map.
**Outputs:** $\mathbf{z}_0$;                              // large scene image.

1 **Function** VCJD $(\{\mathbf{x}_t^{(i)}\}, \{\mathbf{c}^{(i)}\})$:
2    **for** $i = 0, \ldots, M-1$ **do**
3       $y^{(i)}, k^{(i)}, \{(y_n^{(i)}, s_n^{(i)})\}_{n=0\cdots N^{(i)}-1} = \mathbf{c}^{(i)}$ ;
4       $\tilde{\mathbf{x}}_{t-1}^{(i)} \leftarrow \mathcal{D}(\mathbf{x}_t, t, y^{(i)}, k^{(i)}, \{(y_n^{(i)}, s_n^{(i)})\}_{n=0\cdots N^{(i)}-1}, \epsilon)$
5    $\mathbf{z}_{t-1} \leftarrow \frac{\sum_i \mathcal{V}_{i\rightarrow\mathbf{z}}(\tilde{\mathbf{x}}_t^{(i)})}{\sum_i m^{(i)}}$**for** $i = 0, \ldots, M-1$ **do**
6       $\mathbf{x}_{t-1}^{(i)} \leftarrow \mathcal{V}_{\mathbf{z}\rightarrow i}(\mathbf{z}_{t-1})$;
7    **return** $\{\mathbf{x}_{t-1}^{(i)}\}$;

8 $\{\mathbf{c}^{(i)}\}_{i=0\cdots M-1} = \{\mathcal{V}_{\mathbf{C}\rightarrow i}(\mathbf{C})\}_{i=0\cdots M-1}$;
9 $\{\mathbf{x}_T^{(i)}\}_{i=0\cdots M-1} = \{\mathcal{V}_{\mathbf{z}\rightarrow i}(\mathbf{z}_T)\}_{i=0\cdots M-1}$;
10 **for** $t = T, \ldots, 1$ **do**
11    $\{\mathbf{x}_{t-1}^{(i)}\} = $ VCJD $(\{\mathbf{x}_t^{(i)}\}, \{\mathbf{c}^{(i)}\})$ ;
12 $\mathbf{z}_0 \leftarrow \frac{\sum_i \mathcal{V}_{i\rightarrow\mathbf{z}}(\tilde{\mathbf{x}}_0^{(i)})}{\sum_i m^{(i)}}$

---

2023; Zhang et al., 2023c) for single-turn and multi-turn models. For efficient usage of resources, the quantized parameter efficient fine-tuning is adopted. Pixel perturbation swaps each pixel in the interpolated image with the existing pixel in $d_{pert} = 1$ with $p_{pert} = 0.05$. See the supplementary for further details on our implementation and experiments and more results.

## 4.1 EVALUATION OF LARGE IMAGE GENERATION FROM DETAILED TEXT

We begin by evaluating our method for large scene generation form detailed text that is combination of our layout generation part and image generation part, which we targeted for as a main task. We thoroughly evaluated our method with the state-of-the-art text-guided large scene generation methods, MultiDiffusion (Bar-Tal et al., 2023) and SyncDiffuison (Lee et al., 2023) which show promising result by using joint diffusion process with the versatility of pretrained diffusion models. For quantitative comparisons and user study, we obtain hierarchical detailed text describing random 100 real-world complex scene images in test set of CrowdCaption dataset (Wang et al., 2022) and several pretrained image understanding models such as image captioning model (Li et al., 2022) and object detection model (Wang et al., 2023). For qulaitative comparisons and user study, we create new detailed description making the above description more complex, applying various artistic styles and appearance on human and objects to demonstrate the versatility of our method.

**Qualitative comparisons.** As represeted in Fig. 4, our method show outstanding quality. Especially, in terns of faithfulness, the baselines show missing objects and incorrect binding while our method doesn't miss the object and show excellent binding. Also, the baselines show excessive duplication of human and objects, which hinder users from controlling the number of objects, while the issue is alleviated in our method. Furthermore, the results from baselines seem awkward and discrete in global context and human instance has bad structure, while ours show natural across the large image.

**Quantitative comparisons.** We measure global CLIP score (Hessel et al., 2021) to measure the faithfulness and the matching performance of the number of human instances ($N_{\text{human}}$ matching) between the text prompt and the generated images to assess the controllablity of the methods. Our method significantly outperforms baselines achieving higher CLIP score as well as higher $N_{\text{human}}$ matching performance as represented in Table 1.

**User study.** We conducted a user study to further evaluate the faithfulness and naturalness using a crowd sourcing. Participants were presented with large-scene images generated by MultiDiffusion (Bar-Tal et al., 2023), SyncDiffusion (Lee et al., 2023) and our DetText2Scene methods. They were then asked to rank the methods with following the guidelines: Rank the images in order of (1) their faithfulness with the text without missing objects and incorrect binding between words and

G) A ornamental flower gardens and destroyed castle, covered with old dirt and moss, grass. Gr1) the two people are in the left side. P1) a man is wearing suit, looking at robot P2) a woman is wearing white dress, looking at robot Gr2) the Large Robot are in the right, middle side. P1) an ancient ruins of a giant robot, made by huge rocks

G) A realistic sculpted castle and stone walls in the rocky mountains. Gr1) a small group of two man are walking in left under side. P1,P2) a small size man is walking. Gr2) a small group of two woman are walking in right under side. P1,P2) a small size woman is walking

G) Under the beautiful deep sea teeming with vibrant corals, colorful, vivid fishes. Gr1) A diver explores a breathtakingly in to the sea, center of the image. P1) a Diver with skin scuber.

G) Some futuristic city and flying ships, in the style of spiritual landscape, meticulously detailed. Gr1) A person next to the futuristic car. P1) a person with futuristic uniform and goggle. O1) a futuristic car.

| (a) MultiDiffusion | (b) SyncDiffusion | (c) DetText2Scene (Ours) |

Figure 4: Qualitative results of large-scene image generation from the detailed text (2560×1920)

Table 1: Quantitative evaluation of large image generation from the detailed text

| Quantitative result | CLIP score↑ | $N_{\text{human}}$ matching | | | | User study | Preference of ours | | |
| | | Prec.↑ | Rec↑ | F1↑ | | | Faithfulness | Naturalness | |
| | | | | | | | | Global | Human |
| MultiDiffusion | 17.400 | 0.320 | 0.985 | 0.483 | | | | | |
| SyncDiffusion | 32.017 | 0.284 | 0.995 | 0.442 | | vs MultiDiffusion | 71.1% | 67.0% | 65.7% |
| **Ours** | 32.097 | 0.764 | 0.930 | 0.839 | | vs SyncDiffusion | 73.7% | 77.6% | 74.9% |

objects, (2) their naturalness from global context, and (3) their naturalness from a physical structure perspective. As shown in Table 1, out of the 5,490 responses from 122 participants, the results from our method is preferred in term of faithfulness with the text prompt, naturalness from global context and from human physical structure perspective compared to the baselines.

G) A group of motorcyclists standing in front of a mountain. Gr1) a group of people are standing at a distance . they are standing in a line. P1) a man with his arm raised P2) a man in a black jacket P3) a man riding a motorcycle P4) a man standing on the beach P5) a man standing in the distance P6) a man in a white shirt P7) a man standing in the snow P8) a man watching an event

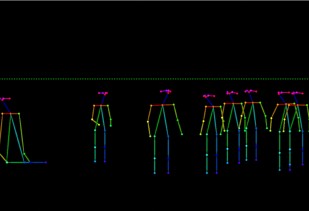 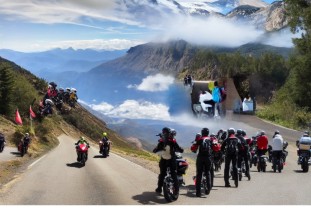 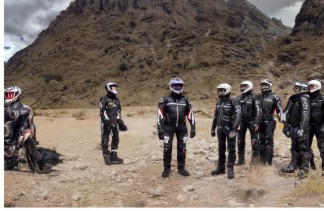

G) A group of people riding horses down a dirt road. Gr1) there are three riders in front . there is a man driving a carriage behind them. P1,P2) a man riding a horse P3) a man in a black jacket O1) a dark horse O2) a white horse O3) dark horse leading other horses

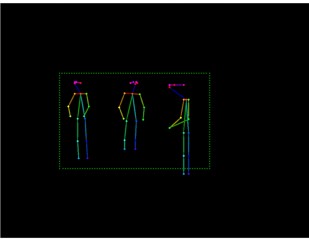 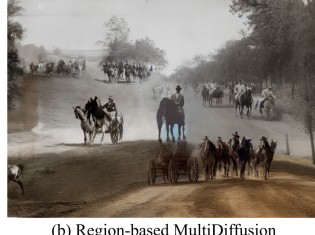 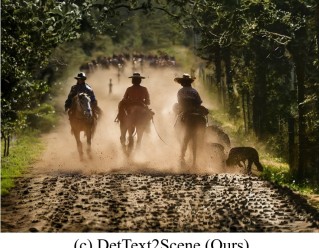

(a) Input keypoint-box layout map      (b) Region-based MultiDiffusion      (c) DetText2Scene (Ours)

Figure 5: Qualitative evaluation of large image generation from the detailed text (2560×1920)

"A group of people riding skis down a snow-covered slope. People are standing, all wear thick coats and hats, a man wearing blue jacket, a man in a red jacket, a man wearing a red jacket, a yellow and black skis. Some people are standing, a person wearing a coat, a person in a red jacket, a person wearing a red jacket."

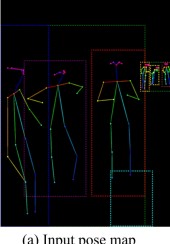 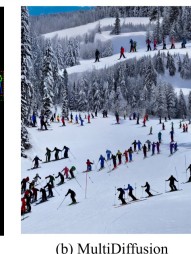 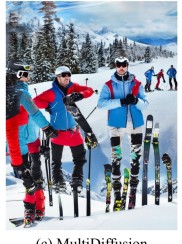 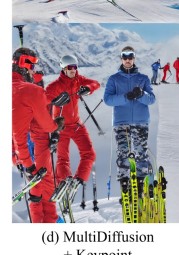 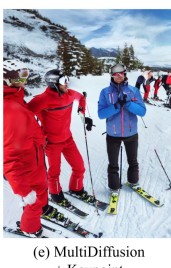

(a) Input pose map    (b) MultiDiffusion    (c) MultiDiffusion + Keypoint    (d) MultiDiffusion + Keypoint + Box layout    (e) MultiDiffusion + Keypoint + Box layout + PPHE

Figure 6: Ablation studies (960×1280)

Table 2: Quantitative evaluation of keypoint-box layout generation part

| | Numerical matching | | | | Group box location acc.↑ | Group box inclusion acc.↑ |
|---|---|---|---|---|---|---|
| | Prec.↑ | Rec.↑ | F1 ↑ | | | |
| $N_{group}$ | 1.000 | 1.000 | 1.000 | Left/right | 0.964 | 0.800 |
| $N_{human}$ | 1.000 | 0.981 | 0.991 | Top/bottom | 0.693 | |
| $N_{obj}$ | 1.000 | 0.911 | 0.954 | Total | 0.600 | |

Table 3: Quantitative evaluation of image generation part. R-MultiDiffusion stands for Region-based Multidiffuison (Bar-Tal et al., 2023).

| Quantitative result | CLIP score↑ | $N_{human}$ matching | | |
|---|---|---|---|---|
| | | Prec.↑ | Rec↑ | F1↑ |
| R-MultiDiffusion | 20.045 | 0.430 | 0.986 | 0.599 |
| **Ours** | 27.220 | 0.877 | 0.927 | 0.901 |

| User study | Preference of ours | | |
|---|---|---|---|
| | Faithfulness | Naturalness | |
| | | Global | Human |
| vs R-MultiDiffusion | 52.6% | 85.0% | 84.2% |

## 4.2 EVALUATION OF KEYPOINT-BOX LAYOUT GENERATION PART

As our method is the first method for predicting the keypoint and box layouts simultaneously from the detailed description, we thoroughly evaluated our method on the test set of CrowdCaption dataset(Wang et al., 2022).

**Quantitative comparisons.** As shown in the left of Table 2, the numerical matching performance of the number of group, human and object instances ($N_{group}$, $N_{human}$ and $N_{obj}$ matching) between the text prompt and the generated keypoint-box layout. We achieved high numerical matching

performance over 0.91 in all cases. As shown in middle and right of the Table. 2, we measured the spatial matching performance between desired group box location and the center of generated group box layout.Moreover, we measured the performance of group box inclusion checking that the generated keypoint is located in the corresponding group box. We demonstrate the credible spatial quality performance except for generating group box located in top/bottom of the image.

## 4.3 EVALUATION OF LARGE IMAGE GENERATION PART

We thoroughly evaluated our method with region-based MultiDiffusion (Bar-Tal et al., 2023) which show feasiblity of utilizing spatial control for large scene generation with joint diffusion process with the versatility of pretrained diffusion models. For quantitative comparisons and user study, we obtain hierarchical detailed text describing random 100 real-world complex scene images in test set of CrowdCaption dataset (Wang et al., 2022) and several pretrained image understanding models such as image captioning model (Li et al., 2022), dense captioning model (Wu et al., 2022), and human pose prediction model (Xu et al., 2022). We choose random 100 images of CrowdCaption testset that is filtered to have total number of humans and objects are 5 to 15 and have size smaller than 1500 pixels.

**Qualitative comparisons.**    As represeted in Fig. 5, our method show outstanding quality. Especially, in terns of faithfulness, the baselines show missing objects and incorrect binding while our method doesn't miss the object and show excellent binding. Also, the baselines show excessive duplication of human and objects, which hinder usuers from controlling the number of objects, while the issue is alleviated in our method. Furthermore, the results from baselines seem awkward and discrete in global context and human instance has bad structure, while ours show natural across the large image.

**Quantitative comparison.**    We use same global CLIP score (Hessel et al., 2021) to assess the faithfulness and $N_{human}$ matching performance to evaluate the controllablity of the methods. Our method significantly outperforms baselines achieving higher CLIP score as well as higher $N_{human}$ matching performance.

**User study.**    Likewise, we conducted a user study to further evaluate the faithfulness and naturalness using a crowd sourcing. As shown in Table 1, out of th 3,660 responses from 122 participants, the results from our method is preferred in term of faithfulness with the text prompt, naturalness from global context and from human physical structure perspective compared to the baselines.

## 4.4 ABLATION STUDIES

Our baseline model, MultiDiffusion, struggle to generate high qulaity of images, producing results with low faithfullness, controllablity, naturalness, By adding keypoint information, the size of human instance become diverse and the quality of pysical structure in human instance is improved as in Fig. 6(b). By adding box groudning, the method resolve issues of missing object and incorrect binding as in Fig. 6(c). By adding pixel purtubation-based enhancement, the image show higher naturalness in global context without disconnectivity as in Fig. 6(d).

## 5 DISCUSSION

**Limitation.**    Although LLM shows prompising results for generating keypoint-box layout, the capability of understanding visual context may be limited. For example, 3D information such as depth information is difficult to understand for LLM. The qulaity of generated large-scene results depends on the power of text-to-image diffusion models. The limitation of the Stable diffusion 1.5 (Rombach et al., 2022) includes falling short of achieving 1) complete photorealism, 2) compositionality, 3) proper face generation, 4) generating images with other languages except English and so on.

**Conclusion.**    We propose DetText2Scene, a novel detailed-text-driven large-scale image synthesis 1) a hierarchical keypoint-box layout conversion from the detailed text 2) a view-wise conditioned joint diffusion process 3) a pixel perturbation-based hierarchical enhancemet. Our method significantly outperforms existing method qualitatively and quantitatively with high faithfulness, high controllability with high naturalness in global context for the given descriptions.

ETHICS STATEMENT

Our method can be applied maliciously to produce visuals that make people feel unpleasant or aggressive. This involves creating imagery that people are likely to find upsetting, frightening, or insulting, as well as information that reinforces stereotypes from the past or present. According to the Stable diffusion Rombach et al. (2022) model card, a misuse of the model includes 1) creating inaccurate, hurtful, or otherwise offensive depictions of individuals, their environment, cultures, religions, 2) intentionally spreading stereotypical portrayals or discriminatory material, 3) impersonating individuals without their consent, 4) sexual content without viewer's permission, 5) depictions of horrifying violence and gore and so on. We thus strongly urge people to use our approach wisely and for the proper intended goals.

REPRODUCIBILITY

We include through implementation details and experimental details with pseudo code in the main paper and supplementary. Code will be released upon publication.

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
