SUPPLEMENTARY

# A IMPLEMENTATION DETAILS

## A.1 HIERARCHICAL KEYPOINT-BOX LAYOUT GENERATION

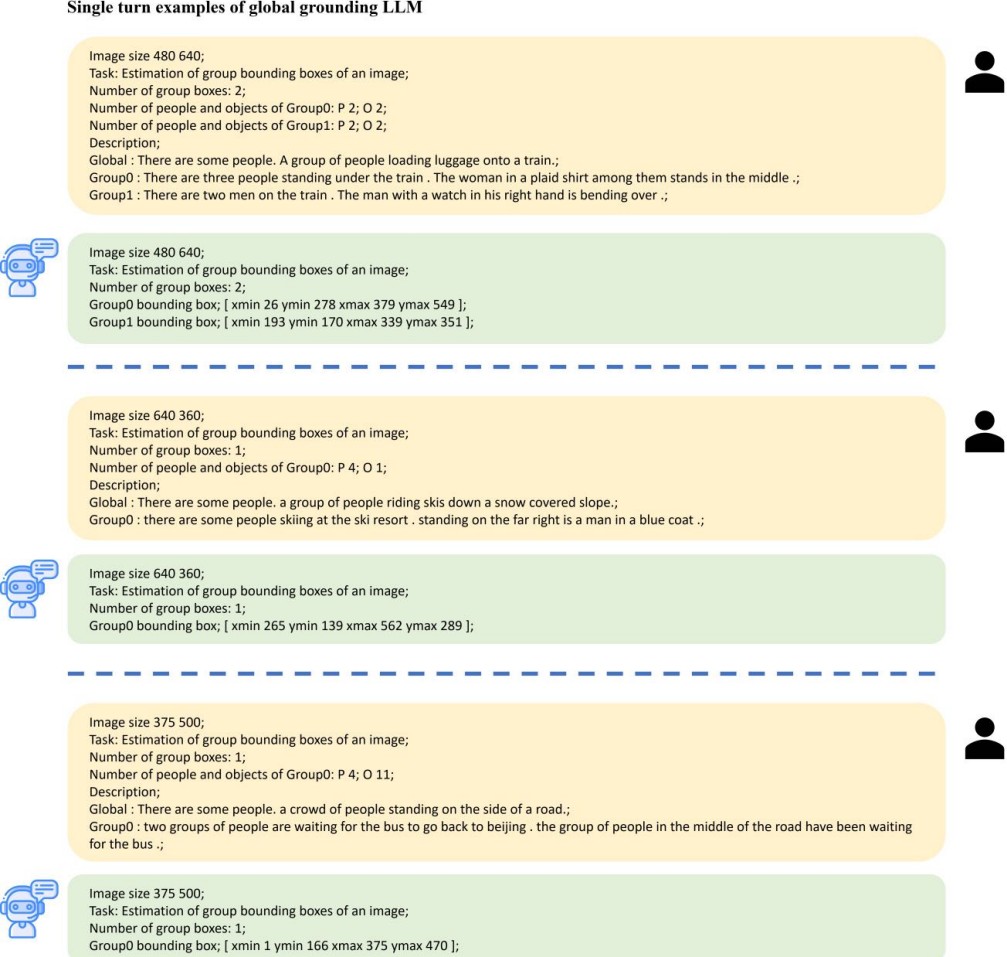

Figure S1: Examples of single turn dialog of global grounding LLM.

Training details of our hierarchical keypoint-box layout generation are presented in the Table. S1. For instruction tuning, we utilized the open-source toolkit of training LLMs (Gao et al., 2023; Zhang et al., 2023c) for single-turn and multi-turn models. For efficient usage of resources, the quantized parameter efficient fine-tuning is adopted. Moreover, we used 4 A100 GPUs and training time was about 5 hours and 1 day for global grounding and local grounding LLM, respectively.

Table S1: Training details of grounding LLM

|  | Model | Annotations | Batch size | Epcoh | Learing rate |
|---|---|---|---|---|---|
| Global grounding LLM | LLAMA2-7B | Group box | 8 | 4 | 3e-5 |
| Local grounding LLM | LLAMA2-7B | Instance keypoint & box | 4 | 4 | 3e-5 |

## A.2 PIXEL PERTURBATION-BASED HIERARCHICAL ENHANCEMENT

**Coarse-to-fine generation.** For Coarse-to-fine generation, we proceed reverse process with $N_{refine}$ times. After each stage of generation, we expanded the height and the width of image $k_{refine}$ times with Pixel

**Multi turn chat example of local grounding LLM**

Image size 640 359;
Task: Estimation of person and object bounding boxes of an image;
Number of group boxes: 1;
Number of people and objects of Group0: P 5; O 1;
Description;
Global : There are some people. a group of people standing on top of a sandy beach.;
Group0 : there is a group of people on the right side of the beach . a man wearing socks stands outside the tent .;
Group0 bounding box; [ xmin 341 ymin 99 xmax 563 ymax 239 ];
Group0;
P0: man wearing a black and yellow wet suit;
P1: man standing under tent;
P2: person in an orange jacket;
P3: man wearing black wet suit;
P4: person in a yellow jacket;
O0: a large white tent;

Image size 640 359;
Task: Estimation of person and object bounding boxes of an image;
Number of people and objects of Group0: P 5; O 1;
P0: [ xmin 341 ymin 104 xmax 380 ymax 236 ];
P1: [ xmin 492 ymin 99 xmax 553 ymax 231 ];
P2: [ xmin 439 ymin 113 xmax 487 ymax 239 ];
P3: [ xmin 397 ymin 133 xmax 424 ymax 215 ];
P4: [ xmin 419 ymin 114 xmax 447 ymax 225 ];
O0: [ xmin 330 ymin 26 xmax 584 ymax 257 ];

Image size 640 359;
Task: Add keypoints of person;
Number of group boxes: 1;
Number of people and objects of Group0: P 5; O 1;
Description;
Global : There are some people. a group of people standing on top of a sandy beach.;
Group0 : there is a group of people on the right side of the beach . a man wearing socks stands outside the tent .;
Group0 bounding box; [ xmin 341 ymin 99 xmax 563 ymax 239 ];
Group0;
P0: man wearing a black and yellow wet suit;
P1: man standing under tent;
P2: person in an orange jacket;
P3: man wearing black wet suit;
P4: person in a yellow jacket;
O0: a large white tent;

Image size 640 359;
Task: Add keypoints of person;
Number of people and objects of Group0: P 5; O 1;
P0: [ person a 374 117 b 359 121 c 365 121 d 363 143 e 364 164 f 352 120 g 348 143 h 352 160 i 357 163 j 356 190 k 347 222 l 350 161 m 352 188 n 350 219 o 375 114 p 377 115 q 370 111 r 365 110 ]; [ xmin 341 ymin 104 xmax 380 ymax 236 ];
P1: [ person a 510 101 b 522 113 c 535 113 d 547 136 e 534 145 f 509 113 g 501 135 h 496 154 i 530 159 j 528 187 k 531 217 l 513 159 m 514 190 n 519 221 o 531 99 p 513 99 q 528 100 r 514 100 ]; [ xmin 492 ymin 99 xmax 553 ymax 231 ];
P2: [ person a 475 125 b 466 136 c 477 136 d 481 156 e 472 153 f 454 136 g 445 156 h 445 170 i 472 172 j 469 203 k 464 231 l 455 172 m 455 203 n 454 231 o 472 121 p 460 121 q 472 122 r 460 122 ]; [ xmin 439 ymin 113 xmax 487 ymax 239 ];
P3: [ person a 407 146 b 412 147 c 403 150 d 401 167 e 403 180 f 421 145 g 424 165 h 417 179 i 413 170 j 413 191 k 415 214 l 423 169 m 422 184 n 407 190 o 405 143 p 409 143 q 403 142 r 414 140 ]; [ xmin 397 ymin 133 xmax 424 ymax 215 ];
P4: [ person a 444 125 b 434 133 c 431 133 d 422 147 e 435 158 f 437 133 g 437 148 h 439 158 i 429 168 j 433 198 k 426 223 l 435 168 m 435 198 n 426 222 o 443 123 p 444 123 q 440 123 r 437 121 ]; [ xmin 419 ymin 114 xmax 447 ymax 225 ];
O0: [ xmin 330 ymin 26 xmax 584 ymax 257 ];

Figure S2: Examples of multi turn chat dialog of local grounding LLM.

perturbation interpolation. For all our qualitative evaluation, we fixed $N_{refine}$ with 3, and for the quantitative evaluation $N_{refine}$ was set to 2. All out experiments were done with $k_{refine} = 2$. Furthermore, except for the first generative process, all the forward process was done with timestep $t_{mix} = 0.5$ with normalized timestep $t \in [0, 1]$.

**Pixel perturbation.** Pixel perturbation swaps each pixel in the interpolated image with the existing pixel in $d_{pert}$ with a certain probability $p_{pert}$. At this time, all experiments were performed using Lanczos interpolation, and $p_{pert} = 0.05$. If $p_{pert}$ is large, the high-frequency component of the interpolated image will be increased and the result after going through the diffusion step will also emphasize high-frequency. If $p_{pert}$ is small, an image in which the low-frequency component is dominant can be obtained. If $p_{pert}$ exceeds a certain level, there is a point where semantic information is lost due to excessive pixel swapping, and artifacts occur. Lanczos resampling Lanczos (1988) was found to be robust for selection of the $p_{pert}$ value because it can inject additional high frequencies while maintaining semantic information because it performs interpolation using the Sinc function. Additionally, the larger $d_{pert}$ is, the wider the pixels are swapped, which results in losing more semantic information and losing the original purpose. Therefore, $d_{pert} = 1$ was decided. The experimental results according to hyper-parameters are presented in the Appendix.

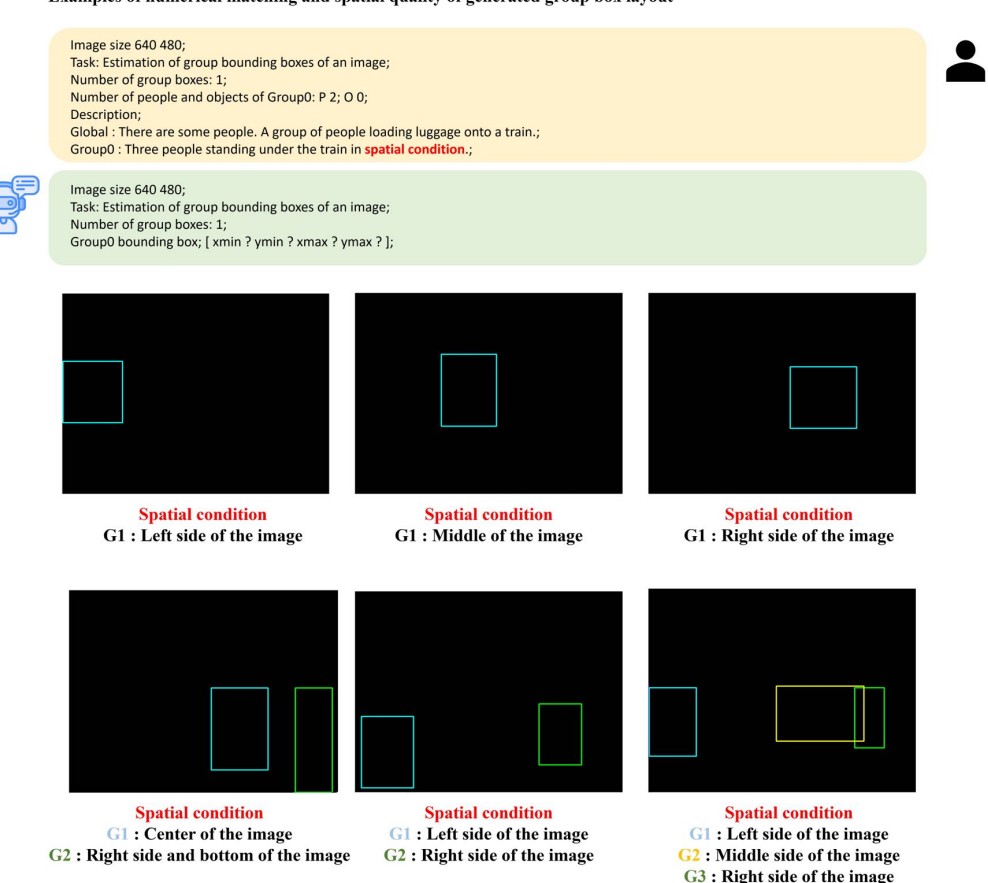

Figure S3: Examples of generated group box layout of global grounding LLM in diverse numerical and spatial cases. The example of single turn dialog is visualized and we varied the number of groups and spatial conditions.

## B    EXPERIMENTAL DETAILS

### B.1    DETAILS ON EVALUATION OF LARGE IMAGE GENERATION FROM DETAILED TEXT

**Data.**    For quantitative comparisons and user study, we obtain hierarchical detailed text describing 100 real-world complex scene images in test set of CrowdCaption dataset (Wang et al., 2022) and several pretrained image understanding models such as image captioning model (Li et al., 2022) and object detection model (Wang et al., 2023). Specifically, we obtain the detailed description describing composed of the global text using pretrained image captioning model (Li et al., 2022), the group text from the annotations in CrowdCpation, and human-object class extracted from pretrained object detection model (Wang et al., 2023) with a template "There are <object1>, <object2>." for constructing natural sentence. For qulaitative comparisons and user study, we create new detailed description making the above description more complex, applying various artistic styles and appearance on human and objects to demonstrate the versatility of our method.

**Quantitative comparison.**    We calculated global CLIP score (Hessel et al., 2021) to measure the faithfulness and the numerical matching performance of the number of human instances ($N_{\text{human}}$ matching) between the text prompt and the generated images to assess the controllablity of the methods. **(1) Global CLIP score:** We calculate the cosine similarity between generated image and corresponding text prompt including global and group description using CLIP-ViT-B/32 model. **(2) $N_{\text{human}}$ matching:** To evaluate the controllability of our proposed DetText2Scene, We measured the numerical matching score and reported the precision, recall and F1 score by following (Feng et al., 2023). We compared the number of human between ground truth from input description and the human counting number of generated image estimated by YOLOv7 (Wang et al., 2023).

**User study.** We conducted a user study to further evaluate the faithfulness and naturalness using a crowd sourcing. Participants were presented with large-scene images generated by MultiDiffusion, SyncDiffusion and our DetText2Scene methods. They were then asked to rank the methods with following the guidelines: Rank the images in order of (1) their faithfulness with the text without missing objects and incorrect binding between words and objects, (2) their naturalness from global context, and (3) their naturalness from a physical structure perspective. The order of the images was shuffled. We crafted detailed caption from random 5 CrowdCaption test images. We placed the results from 3 methods side-by-side. A total of 122 people completed the survey, providing 5,490 votes.

### B.2 DETAILS ON EVALUATION OF KEYPOINT-BOX LAYOUT GENERATION PART

**Quantitative comparisons.** We measure the accuracy of group box, human instances, objects and the accuracy of horizontal and vertical location of the group boxed. As represented in Table 2, our method demonstrate credible performance on both metrics with the score of 80% and 90% .

### B.3 DETAILS ON EVALUATION OF LARGE IMAGE GENERATION PART

**Data.** For quantitative comparisons and user study, we obtain hierarchical detailed text describing random 100 real-world complex scene images in test set of CrowdCaption dataset (Wang et al., 2022) and several pretrained image understanding models such as image captioning model (Li et al., 2022), dense captioning model (Wu et al., 2022), and human pose prediction model (Xu et al., 2022). We choose random 100 images of CrowdCaption testset that is filtered to have total number of humans and objects are 5 to 15 and have size smaller than 1500 pixels as the generation time of region-based MultiDiffuison (Bar-Tal et al., 2023) become longer significantly as the number of objects are increased. Specifically, we obtain the detailed description describing composed of the global text using pretrained image captioning model (Li et al., 2022), the group text from the annotations in CrowdCpation, human-object text extracted from pretrained dense captioning model (Wu et al., 2022), and human keypoints from human pose prediction model (Xu et al., 2022)

**User study.** We conducted a user study to further evaluate the faithfulness and naturalness using a crowd sourcing. Participants were presented with large-scene images generated by region-based MultiDiffusion (Bar-Tal et al., 2023) and our DetText2Scene methods. They were then asked to rank the methods with the same guidelines as used in above. We crafted detailed caption from random 5 CrowdCaption test images. We placed the results from 2 methods side-by-side. A total of 122 people completed the survey, providing 3,660 votes.

## C  ADDITIONAL RESULTS

### C.1  ADDITIONAL RESULTS OF LARGE IMAGE GENERATION FROM DETAILED TEXT

We present additional results of large image generation from detailed text from our method in Fig. S4, Fig. S5, Fig. S6, and Fig. S7.

### C.2  ADDITIONAL RESULTS OF KEYPOINT-BOX LAYOUT GENERATION PART

We present additional results of keypoint-box layout generation part from our method in Fig S8.

### C.3  ADDITIONAL RESULTS OF LARGE IMAGE GENERATION PART

We present additional results of large image generation part from our method in Fig S9 and Fig S10.

**< Text Prompt >**

G) The Legend of Zelda landscape, four girls are dancing on the ground. Gr1) two girls are dancing in the left side. P1) a girl, beautiful game character, wearing a red dress. P2) a girl, beautiful game character, wearing traditional dress. O1) a backpack. Gr2) two girls are dancing in the right side. P1) a girl, beautiful game character, wearing traditional dress. P2) a girl, beautiful game character, wearing a yellow dress.

**<Generated image>**

Figure S4: Qualitative main result of our generated large-scaled scene (2560×1920) from above text prompt.

**< Text Prompt >**

G) A ornamental flower gardens and destroyed castle, covered with old dirt and moss, grass. Gr1) the two people are in the left side. P1) a man is wearing suit, looking at robot P2) a woman is wearing white dress, looking at robot Gr2) the Large Robot are in the right, middle side. P1) an ancient ruins of a giant robot, made by huge rocks

**<Generated image>**

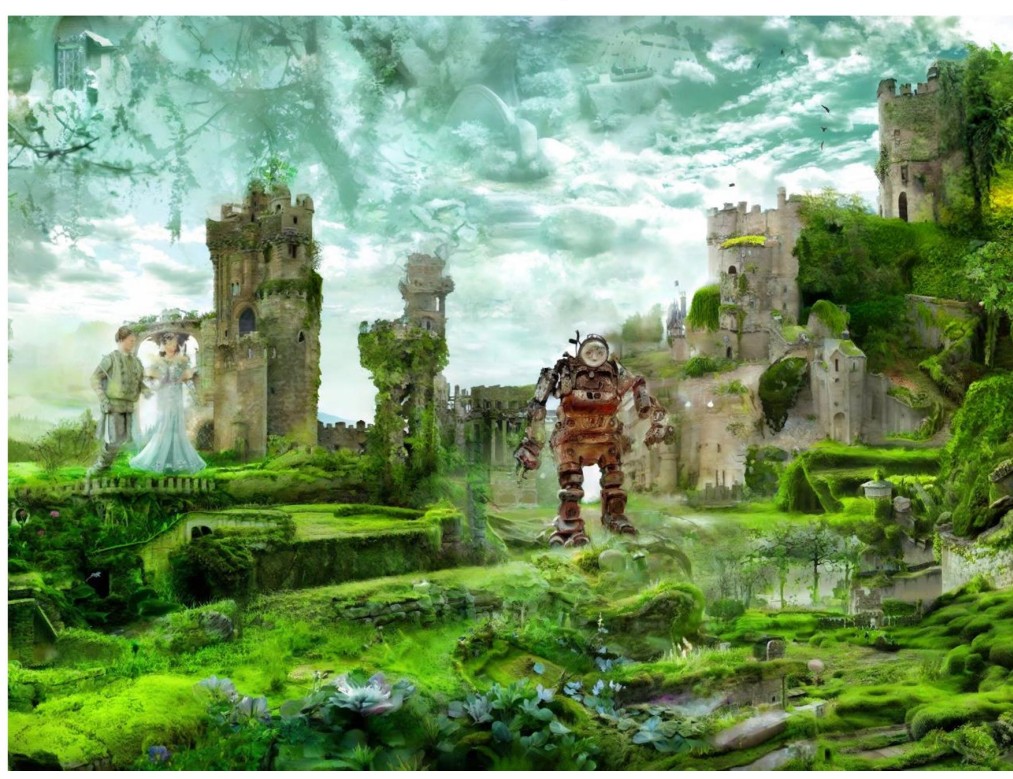

Figure S5: Qualitative main result of our generated large-scaled scene(2560×1920) from above text prompt.

**< Text Prompt >**

G) Under the beautiful deep sea teeming with vibrant corals, colorful, vivid fishes. Gr1) A diver explores a breathtakingly in to the sea, center of the image. P1) a Diver with skin scuber.

**<Generated image>**

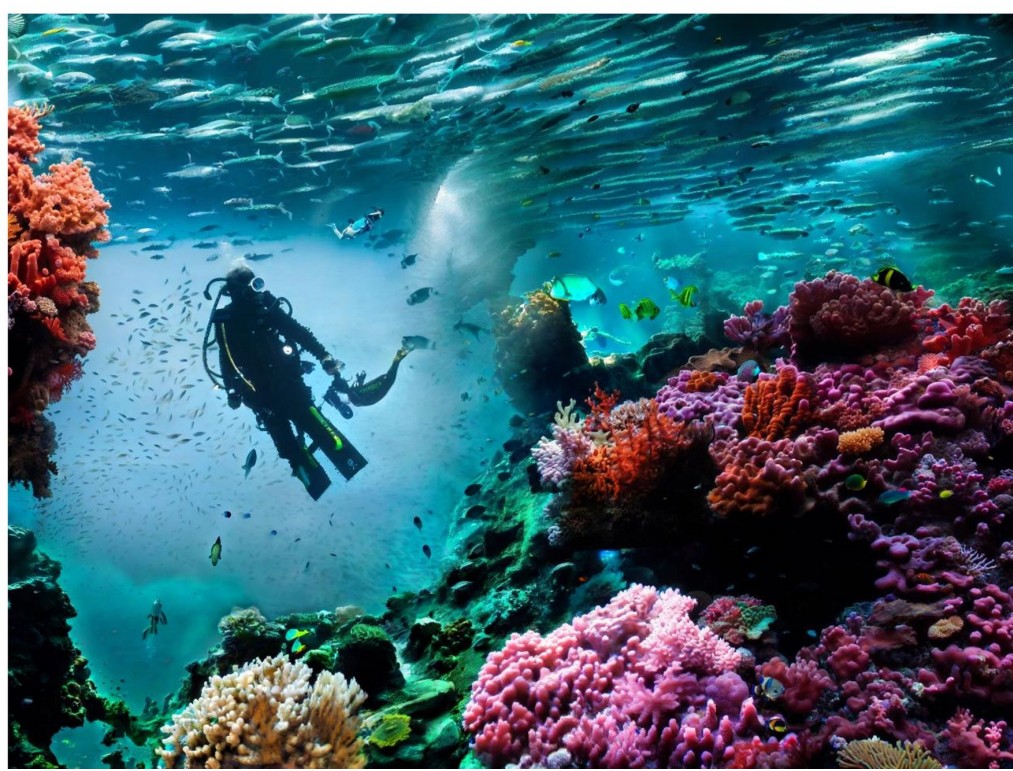

Figure S6: Qualitative main result of our generated large-scaled scene(2560×1920) from text prompt.

**< Text Prompt >**

G) Some futuristic city and flying ships, in the style of spiritual landscape, meticulously detailed. Gr1) A person next to the futuristic car.
P1) a person with futuristic uniform and goggle. O1) a futuristic car.

**<Generated image>**

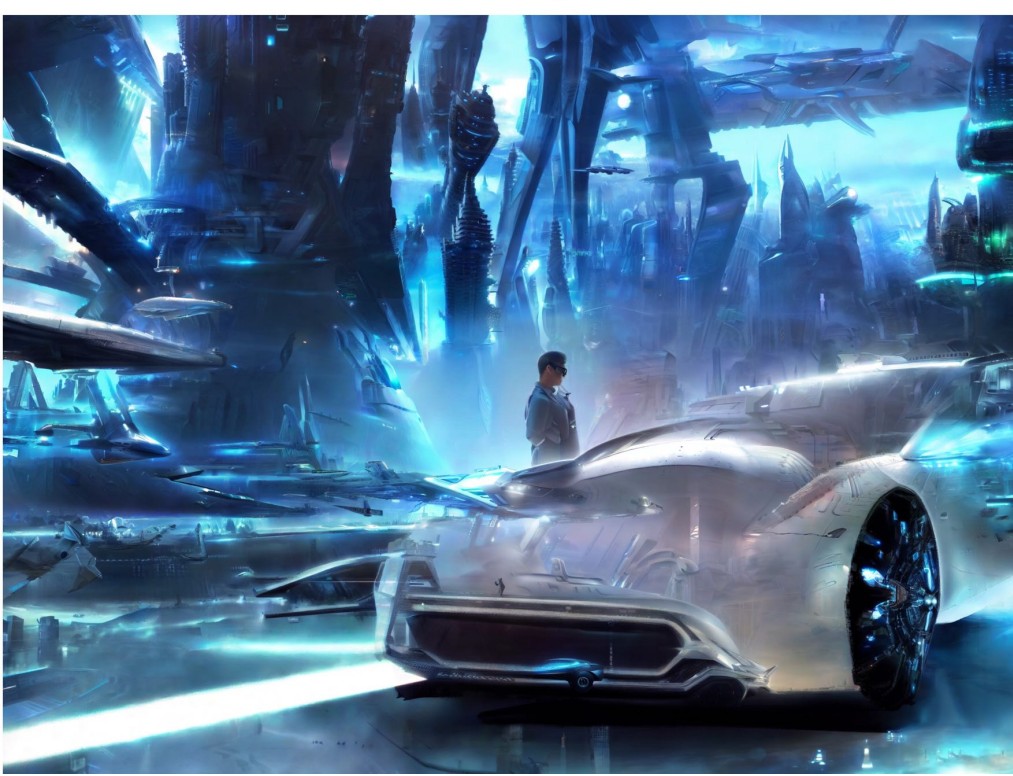

Figure S7: Qualitative main result of our generated large-scaled scene(2560×1920) from above text prompt

**Examples of generated keypoint-box layout**

G) There are some people. a plate with a piece of cake on it. Gr1) a group of girls are sitting around the cake tray . they are sharing the cake. P1) a person is sitting down. P2) a person is sitting down. P3) a person is sitting down. P4) a person is sitting down. P5) a person is sitting down. P6) a person is sitting down. P7) a person is sitting down. O1,O2) a cake. O3) a lightsaber.

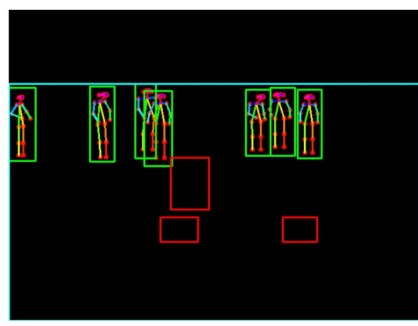

< Text Prompt >                    <Generated keypoint-box layout>

G) There are some people. a group of people getting on a bus. Gr1) two men and a woman are standing on the right side of the door. Both men are dressed in white. P1) woman wearing a purple shirt. P2) man wearing a white cap. P3) a man wearing a white shirt. Gr2) there are three women and a man on the right side of the door . all three women are dressed in black . P1) a woman holding a white bag. P2) woman wearing a blue vest. P3) man wearing black sunglasses. P4) woman wearing black pants. O1) tie..

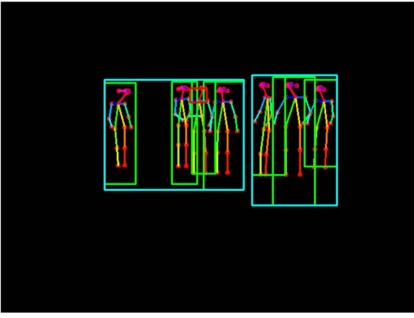

< Text Prompt >                    <Generated keypoint-box layout>

Figure S8: Examples of generated keypoint-box layout from pre-trained global and local grounding LLM from the text prompt. We visualized the keypoint and bounding box of instances and groups.

**< Text Prompt >**

G) A group of motorcyclists standing in front of a mountain. Gr1) a group of people are standing at a distance . they are standing in a line.
P1) a man with his arm raised P2) a man in a black jacket P3) a man riding a motorcycle P4) a man standing on the beach P5) a man standing in the distance P6) a man in a white shirt P7) a man standing in the snow P8) a man watching an event

**<Input keypoint-box layout>**

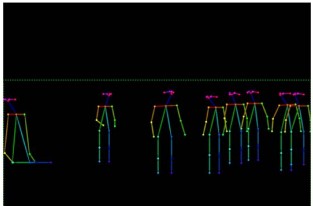

**<Generated image>**

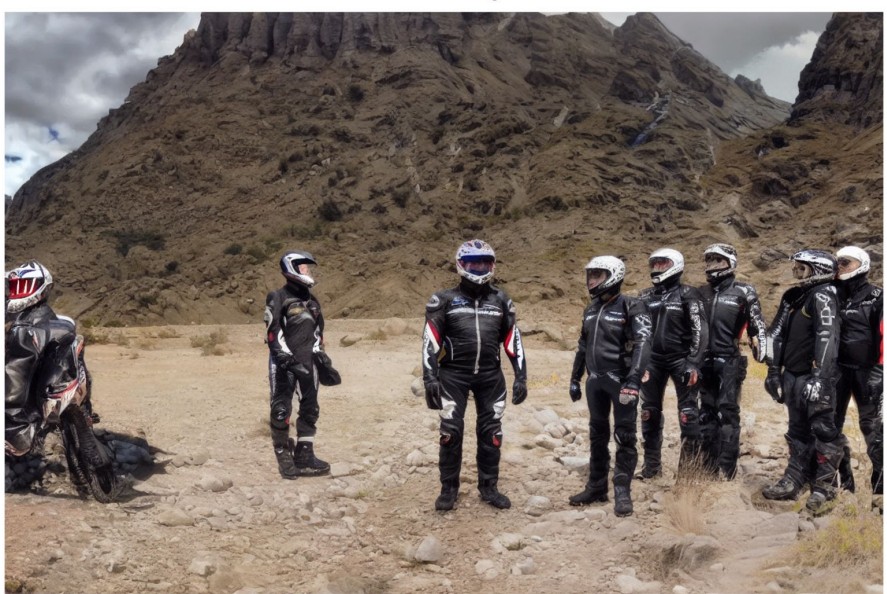

Figure S9: Ablation studies(1280×960)

**< Text Prompt >**

G) A group of people riding horses down a dirt road. Gr1) there are three riders in front . there is a man driving a carriage behind them.
P1,P2) a man riding a horse P3) a man in a black jacket O1) a dark horse O2) a white horse O3) dark horse leading other horses

**<Input keypoint-box layout>**

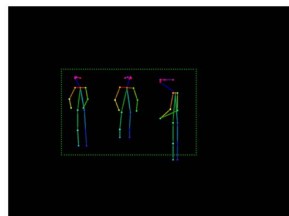

**<Generated image>**

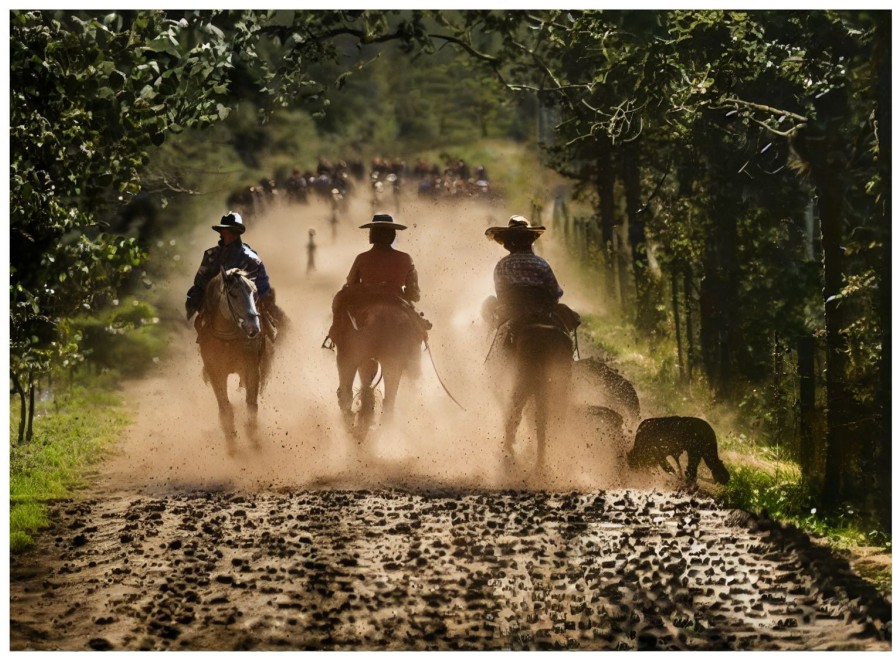

Figure S10: Ablation studies(1280×960)