# OpenReview forum: "Large Scene Synthesis Controlled With Detailed Text Using View-wise Conditional Joint Diffusion With Hierarchical Spatial Controls"
_ICLR.cc/2024/Conference — ICLR 2024 Conference Withdrawn Submission_

### Official Review · Reviewer_F2WC · 2023-10-30

**Soundness:** 2 fair
**Presentation:** 1 poor
**Contribution:** 2 fair
**Rating:** 3
**Confidence:** 4

**Summary:**

The paper proposes a three-stage system that deals with text-to-image generation with long, detailed text conditions. The first stage utilizes finetuned LLMs to generate bounding boxes and human pose keypoints based on global, group, and instance descriptions. The second stage consists of paralleled streams of diffusion process for regional image generation. The third stage refines the consistency of the whole image through multiple rounds of backward-forward diffusion process.

**Strengths:**

- The studied problem is well-motivated and challenging.
- The high-level idea of the proposed method is straightforward and easy to understand.

**Weaknesses:**

Despite the above strengths, the paper has some severe issues in technical details that I am not sure if they are due to the poor writing and presentation of the method.
- Each stage of the system lacks important details for readers to fully understand what's behind the results. For example, I don't understand the aligning process in Sec. 3.1 because there are simply no details about what happens. What are the models used to extract all these annotations? It would be much better to specifically write the model names instead of using a generic name. There are more questions to ask: how did you form the instruction-answer pairs? How many pose keypoints are generated for each group layout? In Sec. 3.2, what specifically is "attention modulation"? I don't think it's a widely known technique/method name that requires a detailed explanation.

- The method seems like a simple combination of multiple previous methods. VCJD seems like a combination of pre-trained ControlNet+"attention modulation" from previous work (Kim et al. 2023b). PPHE seems similar to the superresolution operation proposed in Imagen and other literature.

- The experiments seem not standard and convincing enough. MultiDiffusion is a pretty weak region-guided image generation baseline. Other models like GLIGEN/ReCo/Layout-Guidance should be considered as well. I also feel that there are problems with the evaluation dataset.

Minor:
- Terms are not well defined. For example, what are global, group, and instance descriptions (especially group)? Why would a group description be necessary? It seems that the system assumes all three types of input from human users.

**Questions:**

See weaknesses

---

### Official Review · Reviewer_TMzU · 2023-10-31

**Soundness:** 3 good
**Presentation:** 3 good
**Contribution:** 3 good
**Rating:** 6
**Confidence:** 5

**Summary:**

In this paper, the authors present a novel approach to generating large scenes, providing a detailed analysis of the specific challenges associated with this task. They introduce valuable techniques to address these challenges effectively. Given the paper’s substantial contributions to the field, I believe it warrants acceptance, and I recommend it for publication.

**Strengths:**

The authors have developed a method to generate large scenes from textual descriptions, starting by intelligently deducing keypoints and bounding boxes directly from the language input. This approach seems both clever and logical, demonstrating a thoughtful integration of language understanding and scene generation.


The issue of controllability in large scene generation, especially when incorporating additional text descriptions and dealing with a multitude of objects and human figures, remains underexplored. Addressing this challenge is crucial for advancing the field.



The authors have taken a commendable initiative to address this issue by compiling a dataset tailored to this specific problem.



To infer the bounding box and pose from textual descriptions, the authors employ a language model, a method that is justifiably reasonable and aligns with popular practices in the autonomous driving domain, similar to applications of ChatGPT.

Upon acquiring the bounding box and pose information in textual format, the authors adeptly convert these details into a visual representation. This approach is logical and effectively bridges the gap between linguistic descriptions and visual content.

Leveraging a diffusion model with pose and bounding box conditions is a well-justified approach, especially given my extensive experience in this domain.



Algorithm 1 appears to be both feasible in terms of implementation and reasonable in its approach, indicating a well-thought-out strategy.


The technique of randomly swapping adjacent pixels is a novel and logical approach. I am curious to know if this methodology was initially developed by the authors.

The authors have achieved impressive results.



By analyzing Figures 6d and 6e, it becomes apparent that the technique of randomly swapping pixels plays a significant role. In Figure 6e, the skateboard is correctly positioned under the people’s feet, adhering to logical placement. However, in Figure 6d, the skateboard is misplaced and not under the people’s feet, resulting in a scenario that defies common understanding and expectations. This comparison underscores the effectiveness and importance of the pixel-swapping method.

**Weaknesses:**

The title of this paper appears to be excessively lengthy and somewhat perplexing, which could potentially lead to misunderstandings. I suggest revising it for greater clarity and conciseness.


The technique of randomly swapping adjacent pixels is a novel and logical approach. I am curious to know if this methodology was initially developed by the authors.




In this paper, I deem the ablation study to be of significant importance, warranting a more comprehensive presentation of results and additional figures for a thorough understanding.


There appears to be a technical issue with the PDF (downloaded version) of this paper. Specifically, my computer encounters a crash each time I navigate to the pages containing Figures 2 and 3.  I kindly request the authors to verify and resolve this potential problem to ensure smooth accessibility and review of the material.

**Questions:**

See #Weakness.

---

### Official Review · Reviewer_EaVt · 2023-11-08

**Soundness:** 3 good
**Presentation:** 3 good
**Contribution:** 2 fair
**Rating:** 6
**Confidence:** 4

**Summary:**

This paper proposes a novel method called DetText2Scene for rendering high resolution images with better consistency with respect to conditioning text prompts. To do so, they first leverage a large language model (LLM) to generate spatial keypoint-box layouts from textual descriptions. Secondly, they designed a view-wise conditioned joint diffusion process that synthesizes images conditioned on these layouts. Finally, they found it beneficial to add a pixel perturbation-based hierarchical component, which consists in progressively upsampling and refining the scene for better global consistency and image quality.

**Strengths:**

1. The paper is well written and easy to follow.
2. The idea of using a LLM to generate spatial layouts and provide local guidance for large scene image synthesis is original and interesting.
3. The authors carefully evaluate the layout generation performance and show good performance.
4. The paper demonstrates superior performance of its approach compared to its two main concurrent MultiDiffusion and SyncDiffusion, in terms of quantitative and qualitative metrics as well as user studies.

**Weaknesses:**

1. Figure 3 is difficult to read.
2. The evaluation setting of Table 3 in not clear and i didn’t find where this table was mentioned in the experimental section .
3. The experimental section could be more complete with an evaluation on whether the View-wise conditioned joint diffusion model synthesizes images that are spatially coherent with the bounding boxes (group, human and object masks) on which it is conditioned. Also, there is no quantitative comparison of image quality.
4. Besides, the ablation study is quite limited as it is only done qualitatively by showing one example.
5. Some implementation details are lacking.

**Questions:**

1. Could you increase text font size of Figure 3 so that it’s more readable ? Besides, the drawing describing the view-wise conditioned joint diffusion process could explain better how the U-Net is conditioned on the segments.
2. Could you clarify Table 3 by mentioning it in the experimental section of the main paper? More specifically, how does the experiments in Table 3 differ from Table 1 ? Which region masks did you use to condition R-Multidiffusion?
3. To evaluate the spatial coherence of synthesized images with their conditioning masks, it would be interesting to include mIoU metrics like in MultiDiffusion paper to evaluate the consistency with respect to the generated layouts by passing the synthesized images through a pretrained segmentor.
4. Adding FID metric to measure image quality like it is done in the two main concurrent works MultiDiffusion and Syncdiffusion would strengthen the experimental section.
5. To have a more complete ablation study, I encourage the authors to quantitatively evaluate the benefit of the different components (ie. keypoint, box layout and PPHE) on CLIP score as well as N_human matching performance.
6. About implementation details, could you specify which patch size and sliding window were used to divide the large-scene image in small sub-regions? Is it the same as in MultiDiffusion ? Does it change depending on the enhancement stage of the pixel perturbation-based hierarchical method ? Besides, I didn't find a comparison of the inference time between DetText2Scene and the two other works MultiDiffusion and SyncDiffusion.

---

### Official Review · Reviewer_WVZo · 2023-11-10

**Soundness:** 2 fair
**Presentation:** 1 poor
**Contribution:** 2 fair
**Rating:** 3
**Confidence:** 2

**Summary:**

This paper introduces a multi-step generation method that first generates keypoint-box layout and then uses that to condition on the diffusion process.

**Strengths:**

- It is an interesting approach to use LLMs to generate layout first before generating image

**Weaknesses:**

- Paper is hard to follow. For example, it is unclear what "prompt key-grounded masks" and "prompt dictionary"are and how they are obtained. The subsection on "PIXEL PERTURBATION-BASED HIERARCHICAL ENHANCEMENT" is ambiguous.
- There's a lot of typos in the entire paper. For example,
> "genreation" in Algorithm 1
> "qulaity" in Sec 4.4
> "SyncDiffuison" in Sec 4.1
- The paper's title is on Large Scene Synthesis but the proposed method seems to be targeted at generating better people/humans or humanoid figures on images. There's a serious mismatch between the title/task and proposed method.

**Questions:**

- Why is the segmentation mask in Figure 3(a) different than the one in Figure 3(b)?

**Details Of Ethics Concerns:**

It is heavily focused on human generation. There may be bias coming from LLMs and the paper's model.